# Sobolev Spaces, Kernels and Discrepancies over Hyperspheres

**Simon Hubbert**                                                                  *s.hubbert@bbk.ac.uk*
*Birkbeck, University of London*

**Emilio Porcu**                                                                   *emilio.porcu@ku.ac.ae*
*Khalifa University*
*Trinity College Dublin*

**Chris J. Oates**                                                                 *chris.oates@ncl.ac.uk*
*Newcastle University*
*The Alan Turing Institute*

**Mark Girolami**                                                                  *mag92@eng.cam.ac.uk*
*University of Cambridge*
*The Alan Turing Institute*

**Reviewed on OpenReview:** *https://openreview.net/forum?id=82hRiAbnnm*

## Abstract

This work extends analytical foundations for kernel methods beyond the usual Euclidean manifold. Specifically, we characterise the smoothness of the native spaces (reproducing kernel Hilbert spaces) that are reproduced by geodesically isotropic kernels in the hyperspherical context. Our results are relevant to several areas of machine learning; we focus on their consequences for kernel cubature, determining the rate of convergence of the worst case error, and expanding the applicability of cubature algorithms based on Stein's method. First, we introduce a characterisation of Sobolev spaces on the $d$-dimensional sphere based on the Fourier–Schoenberg sequences associated with a given kernel. Such sequences are hard (if not impossible) to compute analytically on $d$-dimensional spheres, but often feasible over Hilbert spheres, where $d = \infty$. Second, we circumvent this problem by finding a projection operator that allows us to map from Hilbert spheres to finite-dimensional spheres. Our findings are illustrated for selected parametric families of kernel.

## 1 Introduction

This paper precisely characterises the Hilbert spaces reproduced by certain kernels defined over the hypersphere $\mathbb{S}^d$, the $d$-dimensional sphere embedded in $\mathbb{R}^{d+1}$. A kernel $K : \mathbb{S}^d \times \mathbb{S}^d \to \mathbb{R}$ is said to be *geodesically isotropic* if it depends only on the geodesic distance (denoted $\theta$ throughout) between any pair of points located over the spherical shell. Hence, under geodesic isotropy, there exists a mapping $\psi : [0, \pi] \to \mathbb{R}$ such that, for all $\boldsymbol{\xi}, \boldsymbol{\eta} \in \mathbb{S}^d$, $K(\boldsymbol{\xi}, \boldsymbol{\eta}) = \psi(\theta(\boldsymbol{\xi}, \boldsymbol{\eta}))$. Is there a suitable spectral characterisation of Sobolev-type kernels, $K$, under the assumption of geodesic isotropy? The literature is elusive on this aspect.

Several existing contributions to the literature consider the restriction of isotropic kernels from $\mathbb{R}^{d+1}$ to $\mathbb{S}^d$, arguing that under the restriction smoothness properties are retained. This logic is correct, albeit quite unnatural; the restriction of an isotropic kernel in $\mathbb{R}^{d+1}$ to the embedded sphere $\mathbb{S}^d$ has drawbacks that have been emphasised by Gneiting (2013) and by Porcu et al. (2018). For instance, choosing a covariance model which depends on the chordal distance to approximate a random field on the sphere will suffer from poor accuracy at large distances (where the deviation between chordal and geodesic distances are significant). The consequences of this are somewhat difficult to assess theoretically, but intuitively it is unnatural to work

under metrics that are not compatible. To make progress, it becomes necessary to properly define Sobolev spaces for geodesically isotropic kernels over hyperspheres; that is the contribution of this work.

## 1.1 Motivation: Kernels, Cubature, Discrepancies and Beyond

The applications of reproducing kernels are myriad, but the principal motivation for this work was to provide theoretical foundations for the related computational tools of *kernel cubature* and *kernel discrepancy* beyond the usual Euclidean manifold. The simplest and perhaps most useful important of a non-Euclidean manifold is the hypersphere $\mathbb{S}^d$, which is the focus of our discussion, but aspects of the methods developed in this paper are more generally applicable - we return to this point in Section 6.

Let $K : \mathcal{X} \times \mathcal{X} \to \mathbb{R}$ be a reproducing kernel defined on a set $\mathcal{X}$, and let $\mathcal{N}_K$ denote the Hilbert space that is (uniquely) reproduced by this kernel. Let $\delta(x)$ denote a unit mass at $x \in \mathcal{X}$. Given a probability distribution $P$ on $\mathcal{X}$ and a set of locations $\{x_1, \ldots, x_n\} \subset \mathcal{X}$, kernel cubature associates to each location $x_i$ a scalar weight $w_i$, such that the kernel discrepancy

$$D_k \left( \sum_{i=1}^n w_i \delta(x_i), P \right) = \sup_{\|f\|_{\mathcal{N}_K \leq 1}} \left| \sum_{i=1}^n w_i f(x_i) - \int f \, \mathrm{d}P \right| \tag{1}$$

is minimised. These topics have received considerable recent interest in statistics, machine learning, and numerical analysis, where kernel cubature has been applied to such tasks as sampling (Teymur et al., 2021), experimental design (Pronzato and Zhigljavsky, 2020), model selection (Briol et al., 2019), and numerical integration (Jagadeeswaran and Hickernell, 2019). The increasing popularity of kernel cubature is due in part to a closed-form expressions for the cubature weights $w_i$ and to the fact that their error decay is rate-optimal among all cubature methods for $\mathcal{N}_K$, even when the locations $x_i$ are randomly sampled. Further, these tools have received attention in connection with Stein's method from applied probability (Stein, 1972; Gorham and Mackey, 2015), where Chwialkowski et al. (2016); Liu et al. (2016) introduced a kernel discrepancy that can be computed even when $P$ is implicitly defined up to a normalisation constant, addressing a problem that is routinely encountered in the Bayesian context.

Specialising the discussion to $\mathcal{X} = \mathbb{S}^d$, kernel cubature appears in rendering algorithms for glossy surfaces (Marques et al., 2013; 2022) and as a criterion by which the performance of rendering algorithms is measured (Marques et al., 2019), while kernel cubature has been used in combination with Stein's method to numerically approximate posterior expectations in directional statistics (Barp et al., 2022). The theory of kernel cubature on $\mathbb{S}^d$ is well-developed (e.g. as a special case of the general theory of Novak and Woźniakowski, 2008), and optimality properties of kernel cubature have been established in the case where the Hilbert space reproduced by the kernel is equivalent to a Sobolev space on $\mathbb{S}^d$ (Krieg and Sonnleitner, 2021). However, applications of kernel cubature on $\mathbb{S}^d$ are limited by the availability of kernels that satisfy theoretical assumptions and are computationally practical. For example, the methodology of Barp et al. (2022) requires $K$ to reproduce a Hilbert space equivalent to an order-$\beta$ Sobolev space and to admit computable expressions for its derivatives in order for Stein's method to be applied. Further, it is desirable from the perspective of empirical performance for the kernel to be *intrinsically* defined on $\mathbb{S}^d$, to properly reflect the geometry of $\mathbb{S}^d$. If such kernels can be found, then the *Riemann–Stein kernel method* of Barp et al. (2022) facilitates approximation of posterior expectations with error $O(n^{-\beta/d})$, improving on the conventional $O(n^{-1})$ error of Markov chain Monte Carlo whenever $\beta > d/2$ (i.e. when the Sobolev–Hölder embedding is well-defined). An improved understanding of Sobolev-type kernels therefore has the potential to eliminate the gap between the theory and practice of kernel cubature and kernel discrepancy on $\mathbb{S}^d$, and to drive the improvement of methodologies in application areas such as graphics rendering and directional statistics where kernel cubature and kernel discrepancies are used.

Beyond cubature and discrepancies, there is a need for Sobolev kernels within other areas of machine learning. One such example is the use of kernels to approximate the solution of (partial) differential equations (PDEs), where the solution space of the most commonly encountered elliptical PDEs are Sobolev spaces (see e.g. Fasshauer, 2007). The use of a Sobolev kernel then ensures that the solution of the PDE belongs to the RKHS and, through the use of an appropriate kernel method, can be consistently approximated (see Fuselier and Wright, 2009; 2012; Hubbert et al., 2015). The recent interest within the machine learning community

in probabilistic numerics (Hennig et al., 2022), in which kernel methods (Gaussian processes) are used to infer the solution of PDEs, has created a renewed demand for theoretical analyses of the kind reported in this work (e.g. Chen et al., 2021; Krämer et al., 2022; Pförtner et al., 2022). A second important application is in learning with spatial data; kernel methods have been popular since Furrer and Nychka (2007) and are becoming even more popular now, thanks to the fusion of classical spatial statistics with machine learning (Cui et al., 2019; Nikparvar and Thill, 2021). Authors such as Stein (1999) explicitly advocate for the use of Sobolev kernels in this context, as opposed to smoother kernels whose assumptions are unlikely to be satisfied when analysing a real-world dataset.

## 1.2 Contribution

The present article contributes to mathematical understanding of Sobolev-type kernels on $\mathbb{S}^d$, making in particular the following relevant contributions:

**Native and Sobolev Spaces.** We start with the native spaces (reproducing kernel Hilbert spaces) associated with kernels that are geodesically isotropic over $d$-dimensional spheres. We then define suitable Sobolev spaces with exponent $\beta$ for the case of $d$-dimensional spheres. Using Fourier analysis over spheres, we prove that a given kernel, $K$, belongs to a given Sobolev space with exponent $\beta$ if and only if the related Fourier–Schoenberg sequences (see subsequent sections) have a precise rate of decay.

**Hilbert Spheres and Projections.** For $d$ and $K$ given, attaining the Fourier–Schoenberg sequence is extremely difficult. After noting that such sequences are more easily attainable on the (infinite dimensional) Hilbert sphere, we prove that there exists a projection operator that relates those sequences on the Hilbert sphere with their analogue on finite dimensional sphere.

**Euclidean Kernels and Their Restriction.** One potential source of geodesically isotropic kernels, as alluded to above, is to simply restrict a radially isotropic kernel in the ambient Euclidean space $\mathbb{R}^{d+1}$ to the $d-$dimensional sphere. If one knows in advance the Fourier transform of the radial kernel then one can employ a formula due to Narcowich and Ward (2002) to derive the corresponding Fourier–Schoenberg sequence of its restriction to the sphere. We remark that we do not implement the restriction directly (i.e., viewing the problem in $\mathbb{R}^{d+1}$) but instead we take advantage of the fact that the chordal distance can be written in terms of geodesic distance, so that the resulting kernel is geodesically isotropic, thus allowing us to perform a more natural type of harmonic analysis on the sphere.

**Some Parametric Classes of Kernels.** The above ingredients will allow us to prove that celebrated classes of kernels for $d$-dimensional spheres are actually Sobolev kernels, and can be used within the kernel cubature and kernel discrepancy machinery as previously described. A complete Sobolev assessment (derivation of Fourier–Schoenberg sequences complete with precise asymptotic decay rates) for the class of generalised Wendland functions was provided in Hubbert and Jäger (2021), and the present paper adds to this existing knowledge by providing a complete Sobolev assessment for two additional classes, not yet covered in the literature, namely the Matérn class (Stein, 1999) and the $\mathcal{F}$ class Alegría et al. (2021).

The paper proceeds as follows: Section 2 provides a succinct mathematical background. Section 3 starts with expository material on native spaces and then propose a definition of Sobolev space with given exponent. We prove that the Fourier–Schoenberg sequences determine the Sobolev space where the kernel *sits*. Section 4 sets out a framework which delivers closed form expressions for Fourier–Schoenberg sequences derived via the two routes described above, i.e., by projection from the Hilbert sphere and also by restricting a Euclidean radial function to the sphere. We conclude in Section 5 by presenting three explicit parametric cases. Using the tools developed in Section 4 we give closed form expressions for the Fourier–Schoenberg coefficients, we also provide their asymptotic rates of decay and hence conclude by specifying their corresponding Sobolev spaces. A short discussion in Section 6 closes the paper.

## 2   Background

Let $d$ be a positive integer. We consider the $d$-dimensional unit sphere $\mathbb{S}^d = \{\boldsymbol{\xi} \in \mathbb{R}^{d+1}, \|\boldsymbol{\xi}\| = 1\}$, embedded in $\mathbb{R}^{d+1}$, with $\| \cdot \|$ denoting Euclidean norm. We shall also refer to the Hilbert sphere $\mathbb{S}^\infty = \{\boldsymbol{\xi} \in \mathbb{R}^\mathbb{N} : \|\boldsymbol{\xi}\| = 1\}$. We equip $\mathbb{S}^d$ with the great circle (geodesic) distance, defined as $\theta(\boldsymbol{\xi}, \boldsymbol{\eta}) = \arccos(\boldsymbol{\xi}^\top \boldsymbol{\eta}) \in [0, \pi]$, for $\boldsymbol{\xi}, \boldsymbol{\eta} \in \mathbb{S}^d$, where $\top$ denotes transpose. A mapping $K : \mathbb{S}^d \times \mathbb{S}^d \to \mathbb{R}$ is called a kernel if it is positive definite, that is $\sum_{i,j=1}^N a_i a_k K(\boldsymbol{\xi}_i, \boldsymbol{\xi}_j) \geq 0$, for all finite system $\{\boldsymbol{\xi}_i\}_{i=1}^N \subset \mathbb{S}^d$ and constants $a_1, \dots a_N \in \mathbb{R}$. This paper works with geodesically isotropic kernels, so that $K(\boldsymbol{\xi}, \boldsymbol{\eta}) = \sigma^2 \psi(\theta(\boldsymbol{\xi}, \boldsymbol{\eta}))$ for some continuous function $\psi : [0, \pi] \to \mathbb{R}$ with $\psi(0) = 1$, and for $\sigma^2 > 0$. We define $\Psi_d$ as the class of continuous functions $\psi$ being the isotropic part of a kernel $K$ in $\mathbb{S}^d$. We also define $\Psi_\infty = \bigcap_{d=1}^\infty \Psi_d$, with the inclusion relation

$$\Psi_1 \supset \Psi_2 \supset \cdots \supset \Psi_d \supset \cdots \supset \Psi_\infty, \tag{2}$$

being strict. Schoenberg (1942) showed that a continuous mapping $\psi : [0, \pi] \to \mathbb{R}$ belongs to the class $\Psi_d$ if and only if it can be uniquely written as

$$\psi(\theta) = \sum_{m=0}^\infty b_{m,d} \frac{P_m^{(d-1)/2}(\cos(\theta))}{P_m^{(d-1)/2}(1)}, \qquad \theta \in [0, \pi], \tag{3}$$

where $P_m^\lambda$ denotes the $\lambda$-Gegenbauer polynomial of degree $m$ (Abramowitz and Stegun, 1965, 22.2.3), and $\{b_{m,d}\}_{n=0}^\infty$ is a probability mass sequence. Schoenberg (1942) also showed that $\psi$ belongs to the class $\Psi_\infty$ if and only if

$$\psi(\theta) = \sum_{m=0}^\infty b_m (\cos(\theta))^m, \qquad \theta \in [0, \pi], \tag{4}$$

with $\{b_m\}_{m=0}^\infty$ being again a probability mass sequence. We follow Daley and Porcu (2013) and call the sequence $\{b_{m,d}\}_{m=0}^\infty$ in (3) a $d$-Schoenberg sequence. Analogously, we call $\{b_m\}_{m=0}^\infty$ a Schoenberg sequence. Throughout, for a given $d$ and a given element $\psi \in \Psi_\infty \subset \Psi_d$, we call $(\{b_m\}_{m=0}^\infty, \psi)$ and $(\{b_{m,d}\}_{m=0}^\infty, \psi)$ a Schoenberg and a $d$-Schoenberg pair, respectively. Section 4 proves that these pairs are related through an operator defined therein.

For $d = 1$, it is true that (see Gneiting, 2013)

$$b_{0,1} = \int_0^\pi \psi(\theta) \mathrm{d}\theta \qquad \text{and} \qquad b_{m,1} = \frac{2}{\pi} \int_0^\pi \cos(m\theta) \psi(\theta) \mathrm{d}\theta, \text{ for } m \geq 1, \tag{5}$$

while for $d \geq 2$ we have

$$b_{m,d} = \frac{2m + d - 1}{2^{3-d}\pi} \frac{\left(\Gamma\left(\frac{d-1}{2}\right)\right)^2}{\Gamma(d-1)} \int_0^\pi \psi(\theta) P_m^{(d-1)/2}(\cos(\theta)) (\sin\theta)^{d-1} \mathrm{d}\theta. \tag{6}$$

where $\Gamma(\cdot)$ denotes the gamma function (Abramowitz and Stegun, 1965, 6.1.1).

The property of strict positive definiteness is described here through the members $\psi$ of the classes $\Psi_d$. By strict we mean that the inequality in the definition positive definiteness becomes strict provided the real numbers $c_1, \dots, c_n$ are not all zero; we let $\Psi_d^+ \subset \Psi_d$ denote the class of continuous functions $\psi$ associated with a strictly positive definite kernel on $\mathbb{S}^d$. Arguments in Schoenberg (1942) prove that if the elements of the $d$-Schoenberg sequence $\{b_{m,d}\}_{m=0}^\infty$ in (3) are positive for all $m \geq 0$ then $\psi \in \Psi_d^+$. This simple condition is sufficient for our purposes but the reader may consult (D. Chen and Sun, 2003) for a careful investigation of the necessary *and* sufficient conditions.

### 2.1   Harmonic Analysis on Spheres

We now consider members $\psi$ from $\Psi_d^+$ and invoke arguments in Hubbert et al. (2015) to dig into an alternative view of the expansion (3). Specifically, we resort to Fourier expansion through spherical harmonics, that is

$$\psi(\boldsymbol{\xi}^T \boldsymbol{\eta}) = \sum_{m=0}^\infty \sum_{n=1}^{N_{m,d}} \widehat{\psi}_m \mathcal{Y}_{m,n}(\boldsymbol{\xi}) \mathcal{Y}_{m,n}(\boldsymbol{\eta}), \quad \boldsymbol{\xi}, \boldsymbol{\eta} \in \mathbb{S}^d, \tag{7}$$

where $\{\mathcal{Y}_{m,n} : n = 1, \ldots, N_{m,d}\}$ is a real orthonormal basis for the space of spherical harmonics of degree $m$ and the collection $\{\mathcal{Y}_{m,n} : n = 1, \ldots, N_{m,d}, m \geq 0\}$ forms a real orthonormal basis for $L_2(\mathbb{S}^d)$. In addition, $\{\widehat{\psi}_m\}_{m=0}^{\infty}$ is referred to as the sequence of spherical Fourier coefficients for $\psi \in \Psi_d^+$ and these are related to the aforementioned $d$-Schoenberg coefficients via the formula (cf Hubbert et al., 2015, Equation 1.33)

$$b_{m,d} = \frac{\Gamma\left(\frac{d+1}{2}\right) N_{m,d} \widehat{\psi}_m}{2\pi^{\frac{d+1}{2}}}, \tag{8}$$

where $N_{m,d}$ denotes the dimension of the space of spherical harmonics of degree $m$ given by

$$N_{0,d} = 1 \quad \text{and} \quad N_{m,d} = 2\left(m + \frac{d-1}{2}\right)\frac{(m+d-2)!}{(d-1)!m!}. \tag{9}$$

**Remark 2.1.** *The identity (8) proves that the d-Schoenberg $\{b_{m,d}\}_{m=0}^{\infty}$ and the Fourier $\{\widehat{\psi}_m\}_{m=0}^{\infty}$ sequences are linearly related. This justify the vague terminology Fourier–Schoenberg sequences used in the introduction to this paper.*

The positive spherical Fourier coefficients of $\psi \in \Psi_d^+$ decay at a polynomial rate if there exist positive constants $A_1, A_2$ and $\gamma$ such that

$$\frac{A_1}{(1+m)^{d+\gamma}} \leq \widehat{\psi}_m \leq \frac{A_2}{(1+m)^{d+\gamma}}, \quad m \geq 0. \tag{10}$$

Using Stirling's formula (Abramowitz and Stegun, 1965, 6.1.39), that is

$$\Gamma(az + b) \sim \sqrt{2\pi} e^{-az} (az)^{az+b-\frac{1}{2}}, \tag{11}$$

we deduce that $N_{m,d} \sim \frac{2(m+1)^{d-1}}{(d-1)!}$, from which one can also show that there are positive constants $C_1, C_2$, independent of $m$, such that $C_1(m+1)^{d-1} \leq N_{m,d} \leq C_2(m+1)^{d-1}$, $m \geq 0$. Thus, in view of (8), we note that the decay condition (10) on the spherical Fourier coefficients can be recast in terms of the $d-$Schoenberg sequence; specifically, there exist constants $\mathcal{A}_1, \mathcal{A}_2$ such that

$$\frac{\mathcal{A}_1}{(1+m)^{1+\gamma}} \leq b_{m,d} \leq \frac{\mathcal{A}_2}{(1+m)^{1+\gamma}}, \quad m \geq 0. \tag{12}$$

## 2.2 Special Functions

Hypergeometric functions will feature heavily in the course of this work and so we briefly remind the reader that a general hypergeometric function is defined by

$${}_pF_q\begin{bmatrix} a_1 & \cdots & a_p \\ b_1 & \cdots & b_q \end{bmatrix}; z = \sum_{n=0}^{\infty} \frac{(a_1)_n \cdots (a_p)_n}{(b_1)_n \cdots (b_q)_n} \frac{z^n}{n!}, \tag{13}$$

where

$$(c)_n := c(c+1)\cdots(c+n-1) = \frac{\Gamma(c+n)}{\Gamma(c)}, \quad \text{for } n \geq 1, \tag{14}$$

denotes the Pochhammer symbol, with $(c)_0 := 1$. Throughout, $B$ denotes the Beta function, defined for $x > 0$ and $y > 0$ by

$$B(x,y) = \int_0^1 t^{x-1}(1-t)^{y-1}dt = \frac{\Gamma(x)\Gamma(y)}{\Gamma(x+y)}. \tag{15}$$

## 3 Native Spaces on Spheres

For a given $\psi \in \Psi_d^+$, whose spherical Fourier coefficients we assume to be positive, we define the following subspace of $L_2(\mathbb{S}^d)$:

$$\mathcal{N}_\psi = \left\{ f \in L_2(\mathbb{S}^{d-1}) : \|f\|_\psi^2 = \sum_{m=0}^{\infty} \sum_{n=1}^{N_{n,d}} \frac{|\widehat{f}_{m,n}|^2}{\widehat{\psi}_m} < \infty \right\}, \tag{16}$$

where $\widehat{f}_{m,n}$ denote the expansion coefficients associated to the spherical Fourier series representation

$$f = \sum_{m=0}^{\infty} \sum_{n=1}^{N_{n,d}} \widehat{f}_{m,n} \mathcal{Y}_{m,n}, \quad \text{where} \quad \widehat{f}_{m,n} = \int_{\mathbb{S}^d} f(\boldsymbol{\xi}) \mathcal{Y}_{m,n}(\boldsymbol{\xi}) d\omega_d(\boldsymbol{\xi}).$$

We observe that $\| \cdot \|_\psi$ is a norm induced by the inner-product

$$(f, g)_\psi := \sum_{m=0}^{\infty} \sum_{n=1}^{N_{n,d}} \frac{\widehat{f}_{m,n} \widehat{g}_{m,n}}{\widehat{\psi}_m}. \tag{17}$$

We shall call $\mathcal{N}_\psi$ the Native space induced by $\psi$. We observe that if we consider the function $\psi$ whose spherical Fourier coefficients are given by

$$\widehat{\psi}_m := \frac{1}{(1+m)^{2\gamma}},$$

then the corresponding Native space coincides with the Sobolev space of order $\gamma$, that is

$$W_2^\gamma(\mathbb{S}^d) := \Big\{ f \in L_2(\mathbb{S}^{d-1}) : \|f\|_{W_2^\gamma}^2 = \sum_{m=0}^{\infty} \sum_{n=1}^{N_{n,d}} (1+m)^{2\gamma} |\widehat{f}_{m,n}|^2 < \infty \Big\}.$$

More generally, if the spherical Fourier coefficients of $\psi \in \Psi_d^+$ satisfy the decay condition (10), or equivalently, if its $d-$Scohenberg sequence satisfies (12), then the induced Native space $\mathcal{N}_\psi$ is norm equivalent to the Sobolev space $W_2^\beta(\mathbb{S}^d)$ where $\beta = (d+\gamma)/2$. That is the two spaces agree as sets and the norms are equivalent since

$$\sqrt{A_1} \|f\|_\psi \leq \|f\|_{W_2^\beta} \leq \sqrt{A_2} \|f\|_\psi.$$

In particular, if $\mathcal{N}_\psi$ and $\mathcal{N}_{\psi'}$ are norm-equivalent, then their kernel discrepancies equation 1 define the same topology on the space of probability distributions on $\mathbb{S}^d$. We observe that since $\beta > d/2$ then, as a consequence of the Sobolev embedding theorem, the Native space of $\mathcal{N}_\psi$ is continuously embedded in $C(\mathbb{S}^d)$, the space of continuous functions on $\mathbb{S}^d$, and this implies that $\mathcal{N}_\psi$ is a reproducing kernel Hilbert space. The following result concerning Native spaces is adapted from Levesley and Sun (2005) Proposition 3.1.

**Lemma 3.1.** *Let $\Psi(\boldsymbol{\xi}, \boldsymbol{\eta}) = \psi(\boldsymbol{\xi}^\top \boldsymbol{\eta})$ denote a kernel induced by $\psi \in \Psi_d^+$, having expansion (7) according to a Fourier sequence $\{\widehat{\psi}_m\}_{m=0}^{\infty}$ of strictly positive coefficients. The corresponding Native space $\mathcal{N}_\psi$ (16) is a reproducing kernel Hilbert space with reproducing kernel $\Psi$.*

**Remark 3.1.** *Let $\psi \in \Psi_d^+$ induce a kernel $\Psi$ as in Lemma 3.1. If the spherical Fourier coefficients of $\Psi$ satisfy (10), then $\Psi$ is a reproducing kernel for a space that is norm equivalent to the Sobolev space $W_2^\beta(\mathbb{S}^d)$ where $\beta = \frac{d+\gamma}{2}$.*

For two mappings $f, g : \mathbb{N}_0 \to \mathbb{R}$, we say that $f(n) \sim g(n)$ if and only if

$$\lim_{n\to\infty} \frac{f(n)}{g(n)} = 1. \tag{18}$$

A direct implication of (18) is that there exists positive constants $A_1$ and $A_2$ such that

$$A_1 g(n) \leq f(n) \leq A_2 g(n), \quad n \geq 0.$$

In view of Remark 3.1 we observe that by establishing asymptotic decay rates for various classes of covariance function we will be able to establish which order Sobolev space the covariance kernels are reproducing for.

# 4  Quantifying Smoothness on $d$-dimensional Spheres

In this paper we consider parametric classes of members $\psi$ of the class $\Psi_\infty^+$. We access these via the two approaches described in Section 1. Specifically we will either take $\psi \in \Psi_\infty^+$ as a starting point and consider its projection to $\mathbb{S}^d$, or we will take a positive definite radial kernel in $\mathbb{R}^{d+1}$ as a starting point, and consider its restriction to $\mathbb{S}^d$. In both cases we will derive closed form expressions for the associated $d-$Schoenberg sequences and by examining their asymptotic decay we can quantify the smoothness properties which, in turn, determines whether the induced Native space is norm equivalent to a Sobolev space of a certain order.

## 4.1  Projecting $\Psi_\infty^+$ to $\Psi_d^+$.

Many of the well known parametric classes in numerical analysis and statistics are defined through members of the class $\Psi_\infty^+$, i.e., the Schoenberg sequence $(b_m)_{m=0}^\infty$ is known for the representation (4). This is an obstacle in the case where one wants employ such functions on a finite dimensional sphere, where one requires the $d-$Schoenberg sequence in order to quantify the smoothness of the kernel and consequently to state whether the induced Native space is norm equivalent to a Sobolev space of a certain order. In order to circumvent this we consider the following projection operator which we will show maps the Schoenberg sequence of $\psi \in \Psi_\infty$ to its unique $d-$Schoenberg sequence when viewed as a member of $\Psi_d$. To the best of our knowledge, this contribution is novel.

**Remark 4.1** (The Projection Operator). *We define the operator $\Upsilon_d$ that acts pointwise on the coefficients $b_m$ from a Schoenberg sequence $\{b_m\}_{n=0}^\infty$ through the identity*

$$\forall m \in \mathbb{N}_0, \ \Upsilon_d(b_m) = \frac{\sqrt{\pi}}{2^{m+d-2}\Gamma\left(\frac{d}{2}\right)} \frac{\Gamma(m+d-1)}{m!\Gamma\left(m+\frac{d-1}{2}\right)} \sum_{j=0}^\infty b_{m+2j} \frac{(m+2j)!}{j!2^{2j}\left(m+\frac{d+1}{2}\right)_j} \tag{19}$$

*where $(x)_j = \Gamma(x+j)/\Gamma(x)$ denotes the Pochhammer symbol (Abramowitz and Stegun, 1965, 6.1.22).*

**Proposition 4.1** (Projection Operator). *Let $\Upsilon_d$ be as defined through (19). Then, $\Upsilon_d$ maps $\Psi_\infty$ ($\Psi_\infty^+$) into $\Psi_d$ ($\Psi_d^+$). That is, let $b_{m,d}$ be defined as $b_{m,d} = \Upsilon_d(b_m)$, for $m \in \mathbb{N}$ and for $\{b_m\}_{m=0}^\infty$ a Schoenberg sequence. Then, the sequence $\{b_{m,d}\}_{m=0}^\infty$ is a d-Schoenberg sequence.*

*Proof.* According to Bingham (1973) Lemma 1, the following identity holds

$$(\cos\theta)^m = \frac{m!\Gamma\left(\frac{d-1}{2}\right)}{2^m(d-2)!} \sum_{0 \le 2k \le m} \frac{(m-2k+\frac{d-1}{2})(m-2k+d-2)!}{k!(m-2k)!\Gamma\left(m-k+\frac{d+1}{2}\right)} \frac{P_{m-2k}^{(d-1)/2}(\cos(\theta))}{P_{m-2k}^{(d-1)/2}(1)}. \tag{20}$$

This allows us to deduce that

$$\begin{aligned}
\psi(\theta) &= \sum_{m=0}^\infty b_m(\cos(\theta))^m \\
&= \frac{\Gamma\left(\frac{d-1}{2}\right)}{(d-2)!} \sum_{m=0}^\infty \frac{b_m m!}{2^m} \sum_{0 \le 2k \le m} \frac{(m-2k+\frac{d-1}{2})(m-2k+d-2)!}{k!(m-2k)!\Gamma\left(m-k+\frac{d+1}{2}\right)} \frac{P_{m-2k}^{(d-1)/2}(\cos(\theta))}{P_{m-2k}^{(d-1)/2}(1)}.
\end{aligned} \tag{21}$$

By inspection, the coefficient of $\frac{P_m^{(d-1)/2}(\cos(\theta))}{P_m^{(d-1)/2}(1)}$ is given by

$$b_{m,d} = \frac{\Gamma\left(\frac{d-1}{2}\right)\left(m+\frac{d-1}{2}\right)}{2^m} \frac{(m+d-2)!}{m!(d-2)!} \sum_{j=0}^{\infty} b_{m+2j} \frac{(m+2j)!}{j!2^{2j}\Gamma\left(m+j+\frac{d+1}{2}\right)}$$

$$= \frac{\Gamma\left(\frac{d-1}{2}\right)}{\Gamma(d-1)} \frac{\left(m+\frac{d-1}{2}\right)}{\Gamma\left(m+\frac{d+1}{2}\right)} \frac{1}{2^m} \frac{(m+d-2)!}{m!} \sum_{j=0}^{\infty} b_{m+2j} \frac{(m+2j)!}{j!2^{2j}\left(m+\frac{d+1}{2}\right)_j} \tag{22}$$

$$= \frac{\sqrt{\pi}}{2^{m+d-2}\Gamma\left(\frac{d}{2}\right)} \frac{\Gamma(m+d-1)}{m!\Gamma\left(m+\frac{d-1}{2}\right)} \sum_{j=0}^{\infty} b_{m+2j} \frac{(m+2j)!}{j!2^{2j}\left(m+\frac{d+1}{2}\right)_j},$$

where the final line follows from $\Gamma(x+1) = x\Gamma(x)$ and an application the Gamma function identity (Abramowitz and Stegun, 1965, 6.1.18),

$$\frac{\Gamma(2z)}{\Gamma(z)} = \frac{2^{2z-1}}{\sqrt{\pi}}\Gamma\left(z+\frac{1}{2}\right). \tag{23}$$

$\square$

## 4.2 Restricting Radial Kernels to the Sphere

An alternative source of members of $\Psi_d^+$ can be accessed by choosing a radial kernel $\phi$ that is known to be positive definite on $\mathbb{R}^{d+1}$ and then defining its restriction to the sphere. Specifically, we suppose that $d$ is a fixed space dimension and we take a parametric family $\{\phi(\cdot, \boldsymbol{\lambda}), \boldsymbol{\lambda} \in \Theta \subset \mathbb{R}^p\}$ of radial functions that are positive definite on $\mathbb{R}^{d+1}$. The *chordal distance* on $\mathbb{S}^d$ is connected to the *geodesic distance* via

$$d_{\mathrm{CH}}(\boldsymbol{\xi}, \boldsymbol{\eta}) = \|\boldsymbol{\xi} - \boldsymbol{\eta}\| = \sqrt{2 - 2\cos\left(\theta(\boldsymbol{\xi}, \boldsymbol{\eta})\right)} \qquad \boldsymbol{\xi}, \boldsymbol{\eta} \in \mathbb{S}^d. \tag{24}$$

Using this we define

$$\psi(\theta, \boldsymbol{\lambda}) := \phi\left(\sqrt{2 - 2\cos(\theta)}, \boldsymbol{\lambda}\right), \tag{25}$$

and, by construction, this restricted family belongs to $\Psi_d^+$. A crucial ingredient for computing the $d-$Schoenberg coefficients of the restricted family is prior knowledge of the $d-$dimensional radial Fourier transform of $\phi$.

**Definition 4.1.** *Let $\phi(\cdot)$ denote a continuous real valued function on $[0, \infty)$. The $d-$dimensional radial Fourier transform of $\phi$ is defined by Stein (1999)*

$$\widehat{\phi}(r) = \mathcal{F}_d\phi(r) = r^{-\frac{d-2}{2}} \int_0^{\infty} \phi(t) t^{\frac{d}{2}} J_{\frac{d-2}{2}}(rt)dt, \quad r \geq 0, \tag{26}$$

*where $J_\nu(\cdot)$ denotes the Bessel function of the first kind with order $\nu$. We note that a sufficient condition for $\widehat{\phi}(r)$ to be well-defined is that $\phi(t)t^{d-1}$ is absolutely integrable.*

In this framework the $d-$Schoenberg coefficients associated to members of $\Psi_d^+$ that are defined via (25) is given by the following formula (cf. Narcowich and Ward (2002) Theorem 4.1)

$$b_{m,d} = (2\pi)^{\frac{d+1}{2}} \kappa_{m,d} \int_0^{\infty} t J_{m+\frac{d-1}{2}}^2(t)\widehat{\phi}(t)dt,$$

$$\text{where} \quad \kappa_{m,d} = \frac{\Gamma\left(\frac{d+1}{2}\right)}{2\pi^{\frac{d+1}{2}}\Gamma(d)} \frac{(2m+d-1)(m+d-2)!}{m!}. \tag{27}$$

In the next section we use the results presented here on Hilbert space projections and on spherical restrictions to derive closed form expressions for the $d-$Schoenberg coefficients for different classes of parameterised families belonging to $\Psi_d^+$.

## 5 Parameterised Families and Native Sobolev Spaces

The parameterised families under consideration in this paper have been chosen for their flexibility. In each case, one parameter of the family dictates the order of the Sobolev space reproduced by the kernel. There are other examples of kernels in the literature which are known to reproduce Sobolev spaces, although these tend to be less flexibly parametrised. For instance, in the case of $\mathbb{S}^2$, it is shown in Brauchart et al. (2014), that the Cui and Freeden kernel defined by

$$K_{\mathrm{CF}}(\theta) = 1 + \sum_{m=1}^{\infty} \frac{1}{m(m+1)} \frac{P_m^{(\frac{1}{2})}(\cos(\theta))}{P_m^{(\frac{1}{2})}(1)} = 2 - 2\log\left(1 + \sqrt{\frac{1-\cos(\theta)}{2}}\right),$$

is a reproducing kernel for a space that is norm equivalent to $W_2^{\frac{3}{2}}(\mathbb{S}^2)$. This is easily verified here using the decay rate of the expansion coefficients and Remark 3.1. Further, in the same paper it is also shown that the adjusted distance kernel, defined by

$$K_{\mathrm{dist}}(\theta) = 2\sqrt{\pi} - \sqrt{2 - 2\cos(\theta)},$$

is another reproducing kernel for $W_2^{\frac{3}{2}}(\mathbb{S}^2)$. This provides an example of 'equivalent' kernels whose cubature rules achieve the same order of convergence. Further examples of kernels on hyperspheres are provided in Minh et al. (2006), including the infinitely smooth spherical Gaussian

$$K_{\mathrm{Gauss}}(\theta) = \exp\left(-\frac{2-2\cos(\theta)}{\sigma^2}\right), \quad \sigma > 0.$$

It is shown that the corresponding $d-$Schoenberg sequence of this kernel exhibits exponentially fast decay and, consequently, its reproducing kernel Hilbert space is a rather small space of infinitely differentiable functions; we do not consider such cases in this paper, since infinite smoothness is rarely a suitable assumption for the cubature applications that motivated this work. However, the analytical techniques in this paper could also be applied to analyse smooth kernels of this kind.

For each of the families of geodesically isotropic kernels presented in this section we will provide:

**1.** A closed form expression of their $d-$Schoenberg sequence.

**2.** The asymptotic rate of decay of their $d-$Schoenberg sequence.

**3.** The Native Sobolev space for which the kernels are reproducing for.

### 5.1 The Matérn Class of Functions

For $\nu, \alpha > 0$, the Matérn class of functions are well-known positive definite radial kernels defined as (Stein, 1999)

$$\mathcal{M}_{\nu,\alpha}(r) = \frac{2^{1-\nu}}{\Gamma(\nu)}\left(\frac{r}{\alpha}\right)^{\nu}\mathcal{K}_{\nu}\left(\frac{r}{\alpha}\right), \qquad r \geq 0,$$

with $\mathcal{K}_{\nu}$ a modified Bessel function of the second kind of order $\nu$ (Abramowitz and Stegun, 1965)[9.6.22]. The Matérn class has been especially popular in spatial statistics after Stein (1999). We consider the restriction of this family to the sphere which we define as

$$\psi_{\mathcal{M}}(\theta, \boldsymbol{\lambda}) := \mathcal{M}_{\nu,\alpha}(\sqrt{2-2\cos(\theta)}), \ \text{ for } \boldsymbol{\lambda} = (\nu, \alpha)^{\top} \in [0,\infty)^2. \tag{28}$$

In order to apply (27) and derive the $d-$Schoenberg coefficients associated to the family $\psi_{\mathcal{M}}(\theta, \boldsymbol{\lambda})$ we require prior knowledge of the radial Fourier transform of $\mathcal{M}_{\nu,\alpha}(r)$. This is given in the following result.

**Lemma 5.1.** *Let $\alpha$ and $\nu$ be positive real numbers. The $d-$dimensional radial Fourier transform of the Matérn kernel $\mathcal{M}_{\nu,\alpha}(r)$ is given by*

$$\widehat{\mathcal{M}_{\nu,\alpha}}(r) = \mathcal{F}_d\mathcal{M}_{\nu,\alpha}(r) = \frac{2^{\frac{d}{2}}\Gamma\left(\nu + \frac{d}{2}\right)}{\alpha^{2\nu}\Gamma(\nu)}\frac{1}{\left(\frac{1}{\alpha^2} + r^2\right)^{\nu + \frac{d}{2}}}. \tag{29}$$

A proof of this result can be found in Stein (1999).

Equipped with the expression for $\widehat{\mathcal{M}_{\nu,\alpha}}(r)$ we can now employ (27) to derive the $d-$Schoenberg coefficients and investigate their asymptotic decay rate. This leads us to the following result.

**Proposition 5.2.** *Let $\boldsymbol{\lambda} = (\nu, \alpha)^\top \in [0, \infty)^2$ and consider the spherical Matérn family $\psi_{\mathcal{M}}(\theta, \boldsymbol{\lambda})$ given by (28). Then we have that*

1. *The $d-$Schoenberg coefficients are given by*

$$
\begin{aligned}
b_{m,d,\mathcal{M}}(\boldsymbol{\lambda}) = (2\pi)^{\frac{d}{2}} \frac{2^{\frac{d}{2}} \Gamma\left(\nu + \frac{d}{2}\right)}{\Gamma(\nu)\alpha^{2\nu}} \frac{\Gamma(m-\nu)\kappa_{m,d}}{\Gamma(m+\nu+d)} \,_1F_2\left[\begin{matrix} \nu + \frac{d}{2} \\ \nu+1-m \quad m+\nu+d \end{matrix}; \frac{1}{\alpha^2}\right] \\
+ \frac{2\pi^{\frac{d+3}{2}}}{\Gamma(\nu)} \frac{(-1)^m \kappa_{m,d}}{\Gamma(m+1-\nu)\Gamma\left(m+\frac{d+1}{2}\right)(2\alpha)^m} \,_1F_2\left[\begin{matrix} m + \frac{d}{2} \\ m-\nu+1 \quad 2m+d \end{matrix}; \frac{1}{\alpha^2}\right].
\end{aligned}
$$

(30)

2. *Further,*

$$
b_{m,d,\mathcal{M}}(\boldsymbol{\lambda}) \sim \frac{2}{\alpha^{2\nu}} \frac{\Gamma\left(\nu + \frac{d}{2}\right)}{\Gamma(\nu)\Gamma\left(\frac{d}{2}\right)} \frac{1}{m^{1+2\nu}}.
$$

3. *The native space $\mathcal{N}_{\psi_{\mathcal{M}}}$ associated with the Matérn kernel is a reproducing kernel Hilbert space with reproducing kernel $\Psi_{\mathcal{M}}(\mathbf{x}, \mathbf{y}) = \psi_{\mathcal{M}}(\mathbf{x}^T\mathbf{y}; \boldsymbol{\lambda})$. Furthermore, $\mathcal{N}_{\psi_{\mathcal{M}}}$ is norm equivalent to the Sobolev space $W_2^\beta(\mathbb{S}^d)$ where $\beta = \nu + \frac{d}{2}$.*

*Proof.* See Appendix A. □

## 5.2 The $\mathcal{F}$-Class of Functions

Recently, Alegría et al. (2021) have proposed the $\mathcal{F}$ family by

$$
\psi_{\mathcal{F}}(\theta, \boldsymbol{\lambda}) = \frac{B(\alpha, \nu+\tau)}{B(\alpha, \nu)} \,_2F_1(\tau, \alpha, \alpha+\nu+\tau; \cos(\theta)), \quad \boldsymbol{\lambda} = (\tau, \alpha, \nu)^\top \in (0, \infty)^3,
$$

(31)

where $\theta \in [0, \pi]$, $B$ is the Beta function defined by (15) and $_2F_1$ is defined through (13).

**Proposition 5.3.** *Let $\boldsymbol{\lambda} = \in (\tau, \alpha, \nu)^\top \in \mathbb{R}_+^3$ denote the parameter vector associated with (31). The corresponding Schoenberg sequence is given by*

$$
b_{m,\mathcal{F}}(\boldsymbol{\lambda}) = \frac{B(\alpha, \nu+\tau)}{B(\alpha, \nu)} \frac{(\tau)_m(\alpha)_m}{(\alpha+\nu+\tau)_m m!} > 0 \quad m \geq 0,
$$

(32)

*and consequently $\psi_{\mathcal{F}}(\theta, \boldsymbol{\lambda})$ belongs to the class $\Psi_\infty^+$.*

*Proof.* This follows from the definition of the hypergeometric $_2F_1$ (13). The coefficients are positive since the parameters of $\boldsymbol{\lambda}$ are positive. □

Equipped with the expression for $b_{m,\mathcal{F}}(\boldsymbol{\lambda})$ we can now employ the projection operator (19) to derive the $d-$Schoenberg coefficients and investigate their asymptotic decay rate. This leads us to the following result.

**Proposition 5.4.** *Let*

$$
\left\{ (\{b_{m,\mathcal{F}}(\boldsymbol{\lambda})\}_{m=0}^\infty, \psi_{\mathcal{F}}(\theta, \boldsymbol{\lambda})); \quad \boldsymbol{\lambda} = \in (\tau, \alpha, \nu)^\top \in \mathbb{R}_+^3 \right\}
$$

*be the Schoenberg pair for the $\mathcal{F}-$family as given in Proposition 5.3. Then,*

1. *The d-Schoenberg sequence $\{b_{m,d,\mathcal{F}}(\boldsymbol{\lambda})\}_{m=0}^\infty$ is uniquely determined through*

$$
b_{m,d,\mathcal{F}}(\boldsymbol{\lambda}) = C_{m,d}(\tau, \alpha, \nu) \,_4F_3\left[\begin{matrix} \frac{\alpha+m}{2} \quad \frac{\alpha+m+1}{2} \quad \frac{\tau+m}{2} \quad \frac{\tau+m+1}{2} \\ \frac{\alpha+\nu+\tau+m}{2} \quad \frac{\alpha+\nu+\tau+m+1}{2} \quad m+\frac{d+1}{2} \end{matrix}; 1\right],
$$

(33)

*where*

$$C_{m,d}(\tau, \alpha, \nu) = \frac{b_m(\tau, \alpha, \nu)}{2^{m+d-2}} \frac{\Gamma(m+d-1)}{\Gamma\left(m + \frac{d-1}{2}\right)} \frac{\sqrt{\pi}}{\Gamma\left(\frac{d}{2}\right)}. \tag{34}$$

2. *It is true that*

$$b_{m,d,\mathcal{F}}(\boldsymbol{\lambda}) \sim \frac{\Gamma(\nu + \alpha)\Gamma(\nu + \tau)}{\Gamma(\alpha)\Gamma(\nu)\Gamma(\tau)} \frac{2^{\nu+1}\Gamma\left(\frac{d}{2} + \nu\right)}{\Gamma\left(\frac{d}{2}\right)} \frac{1}{m^{1+2\nu}}. \tag{35}$$

3. *The native space $\mathcal{N}_{\psi_{\mathcal{F}}(\boldsymbol{\lambda})}$ associated with $\mathcal{F}_{\tau,\alpha,\nu} \in \Psi_d^+$ is a reproducing kernel Hilbert space with reproducing kernel $\mathcal{F}_{\tau,\alpha,\nu}(\mathbf{x}^T\mathbf{y})$. Furthermore, $\mathcal{N}_{\psi_{\mathcal{F}}(\boldsymbol{\lambda})}$ is norm equivalent to the Sobolev space $W_2^{\beta}(\mathbb{S}^d)$ where $\beta = \nu + \frac{d}{2}$.*

*Proof.* See Appendix B. $\qquad\square$

### 5.3 The Generalised Wendland Family

The Generalised Wendland family of radial functions are defined as

$$\begin{aligned}
\mathcal{W}_{\nu,\alpha,\epsilon}(r) &= \frac{1}{B(2\alpha, \nu+1)} \int_{\epsilon r}^{1} \mathcal{W}_{\nu,0,1}(t)\, t\, \left(t^2 - (\epsilon r)^2\right)^{\alpha-1} \mathrm{d}t \\
&= \frac{B(\alpha, \nu+1)}{2^{\nu+1}B(2\alpha, \nu+1)} \left(1 - (\epsilon r)^2\right)^{\nu+\alpha} {}_2F_1\left[\begin{matrix} \frac{\nu}{2} & \frac{\nu+1}{2} \\ \nu+\alpha+1 \end{matrix}; 1-(\epsilon r)^2\right] \quad r \in \left[0, \frac{1}{\epsilon}\right],
\end{aligned} \tag{36}$$

where $\nu > 0$, $\alpha > 0$ and the constant multiplier is chosen to ensure $\phi_{\nu,\alpha,\epsilon}(0) = 1$. Here, $\mathcal{W}_{\nu,0,\epsilon}(r) := (1-\epsilon r)_+^{\nu}$, with $(a)_+$ denoting the positive part of the real number $a$. We note that the functions in this family are compactly supported, where the parameter $\epsilon$ controls the size of the supporting interval.

Arguments in Chernih and Hubbert (2014) show that $\mathcal{W}_{\nu,\alpha,\epsilon}(r)$ is positive definite on $\mathbb{R}^{d+1}$ provided that $\nu \geq \frac{d+2}{2} + \alpha$ and so, under these conditions, we can define their restriction to the sphere $\mathbb{S}^d$ via

$$\psi_{\mathcal{W}}(\theta, \boldsymbol{\lambda}) = \mathcal{W}_{\nu,\alpha,\epsilon}(\sqrt{2 - 2\cos(\theta)}), \qquad \boldsymbol{\lambda} = (\nu, \alpha, \epsilon)^\top \in \mathbb{R}_+^3, \tag{37}$$

where $\theta \in [0, \pi]$. By construction $\psi_{\mathcal{W}}(\theta, \boldsymbol{\lambda})$ belong to $\Psi_d^+$ provided $\nu \geq \frac{d+2}{2} + \alpha$. The properties of these restricted functions have been investigated in detail in Hubbert and Jäger (2021) and these findings are summarised in the following result.

**Theorem 5.5.** *Let*

$$\left\{ \left(\{b_{m,d,\mathcal{W}}(\boldsymbol{\lambda})\}_{m=0}^{\infty}, \psi_{\mathcal{W}}(\theta, \boldsymbol{\lambda})\right); \ \boldsymbol{\lambda} = (\alpha, \nu, \epsilon)^\top \in \mathbb{R}_+^3 \right\}$$

*be the $d-$Schoenberg pair for the generalised Wendland family (37). Then,*

1. *It is true that*

$$\begin{aligned}
b_{m,d,\mathcal{W}}(\boldsymbol{\lambda}) &= \frac{2\Gamma(2\alpha + \nu + 1)}{\Gamma(2\alpha + \nu + 1 + d)B\left(\alpha + \frac{1}{2}, \frac{d}{2}\right)} \frac{1}{\epsilon^d} \\
&\times \frac{\left(m + \frac{d-1}{2}\right)(m+d-2)!}{m!} {}_3F_2\left[\begin{matrix} -\left(m + \frac{d-2}{2}\right) & m + \frac{d}{2} & \frac{d+1}{2} + \alpha \\ \frac{d+1}{2} + \alpha + \frac{\nu}{2} & \frac{d+1}{2} + \alpha + \frac{\nu+1}{2} \end{matrix}; \frac{1}{4\epsilon^2}\right].
\end{aligned} \tag{38}$$

2. *There exist two positive constants $\mathcal{A}_1 < \mathcal{A}_2$ such that*

$$\frac{\mathcal{A}_1 \epsilon^{2\alpha+1}}{(1+m)^{2+2\alpha}} \leq b_{m,d,\mathcal{W}}(\boldsymbol{\lambda}) \leq \frac{\mathcal{A}_2 \epsilon^{2\alpha+1}}{(1+m)^{2+2\alpha}}. \tag{39}$$

3. *The native space $\mathcal{N}_{\psi_{\mathcal{W}}}$ associated to $\psi_{\mathcal{W}}$ is a reproducing kernel Hilbert space with reproducing kernel $\psi_{\mathcal{W}}(\mathbf{x}^T\mathbf{y}, \boldsymbol{\lambda})$. Furthermore, $\mathcal{N}_{\psi_{\mathcal{W}}}$ is norm equivalent to the Sobolev space $W_2^{\beta}(\mathbb{S}^d)$ where $\beta = \alpha + \frac{1}{2} + \frac{d}{2}$.*

*Proof.* The expression for the $d-$Schoenberg coefficients of the generalised Wendland functions can be derived from the closed form expression of their spherical Fourier coefficients as computed in Hubbert and Jäger (2021) (Theorem 4.7), together with (8). The tight asymptotic bounds follow from Hubbert and Jäger (2021) (Theorem 5.8). Lemma 3.1 shows that the native space $\mathcal{N}_{\psi_{\mathcal{W}}}$ possesses the stated reproducing kernel properties. The norm equivalence of $\mathcal{N}_{\psi_{\mathcal{W}}}$ to the Sobolev space of order $\alpha + \frac{1}{2} + \frac{d}{2}$ follows from Remark 3.1 and the decay condition (39) on $b_{m,d,\mathcal{W}}(\boldsymbol{\lambda})$. $\square$

## 6 Discussion

This paper provides new tools that allow for a precise identification of the Sobolev space associated with a given kernel defined over a $d$-dimensional hypersphere. An immediate consequence of our results is an improved understanding of kernel cubature, since once a Sobolev space associated to a kernel has been identified one can determine the rate of convergence of the associated discrepancy (i.e. the worst-case cubature error), using for example the techniques described in Krieg and Sonnleitner (2021) and the references therein. Our results also extend the applicability of the Riemann–Stein cubature method of Barp et al. (2022), used to accelerate posterior computation in the Bayesian context, since this method requires the Sobolev space associated with a kernel to be precisely identified.

Some further extensions of our results might be possible at the expense of additional effort. For instance, we are confident that the extension of the present work to the case of compact two-point homogeneous spaces would apply *mutatis mutandis* by replacing the Gegenbauer polynomials in the Schoenberg expansion with Jacobi polynomials. Some other extensions might be more challenging. For instance, we are unaware at the moment of how to characterise Sobolev cases on hyperspheres when the kernels is not isotropic, but axially symmetric only (Jones, 1963). Another interesting case would be that of product spaces involing the hypersphere with any locally compact group. Finally, we would like to mention that the recent *tour de force* by Wynne et al. (2022) opens for considering the present work in the direction of operator valued kernels.

## Acknowledgement

EP was supported by FSU-2021-016 at Khalifa University. CJO was supported by EPSRC [EP/W019590/1]. MG was supported by a Royal Academy of Engineering Research Chair and EPSRC [EP/T000414/1, EP/R018413/2, EP/P020720/2, EP/R034710/1, EP/R004889/1].

## A Results Associated to the Matérn Kernel

Here we present the proof of the 3 statements of Proposition 5.2 associated to the Matérn kernel.

**Proposition 5.2 Statement 1.**

*Proof.* Using (27) and (29) we can write

$$
\begin{aligned}
b_{m,d,\mathcal{M}}(\boldsymbol{\lambda}) &= (2\pi)^{\frac{d+1}{2}} \kappa_{m,d} \int_0^\infty t J_{m+\frac{d-1}{2}}^2(t) \mathcal{F}_{d+1} \mathcal{M}_{\nu,\alpha}(t) dt \\
&= (2\pi)^{\frac{d+1}{2}} \kappa_{m,d} \frac{2^{\frac{d+1}{2}} \alpha^{d+1} \Gamma\left(\nu + \frac{d+1}{2}\right)}{\Gamma(\nu)} \int_0^\infty \frac{t J_{m+\frac{d-1}{2}}^2(t)}{(1+\alpha^2 t^2)^{\nu+\frac{d+1}{2}}} dt \\
&= \pi^{\frac{d+1}{2}} \kappa_{m,d} \frac{2^{d+1} \Gamma\left(\nu + \frac{d+1}{2}\right)}{\Gamma(\nu) \alpha^{2\nu}} \int_0^\infty \frac{t J_{m+\frac{d-1}{2}}^2(t)}{\left(\frac{1}{\alpha^2} + t^2\right)^{\nu+\frac{d+1}{2}}} dt.
\end{aligned}
\tag{40}
$$

The following formula is adapted from Prudnikov et al. (1981b) 2.12.32.10

$$\int_0^\infty \frac{t^{\beta-1} J_\mu^2(t)}{(z^2+t^2)^\rho} dt$$

$$= \frac{1}{2^{2\rho+1-\beta}} \frac{\Gamma\left(\mu-\rho+\frac{\beta}{2}\right)\Gamma(1+2\rho-\beta)}{\Gamma\left(\rho+1-\frac{\beta}{2}\right)^2 \Gamma\left(\mu+\rho+1-\frac{\beta}{2}\right)} {}_2F_3\left[\begin{array}{c} \rho+\frac{1-\beta}{2}\ \ \rho \\ \rho+1-\mu-\frac{\beta}{2}\ \ \rho+1+\mu-\frac{\beta}{2}\ \ 1+\rho-\frac{\beta}{2} \end{array}; z^2\right]$$

$$+ \frac{z^{2\mu+\beta-2\rho}}{2^{2\mu+1}} \frac{\Gamma\left(\rho-\mu-\frac{\beta}{2}\right)\Gamma\left(\mu+\frac{\beta}{2}\right)}{\Gamma(\rho)\Gamma(\mu+1)^2} {}_2F_3\left[\begin{array}{c} \mu+\frac{1}{2}\ \ \frac{\beta}{2}+\mu \\ 1-\rho+\frac{\beta}{2}+\mu\ \ \mu+1\ \ 2\mu+1 \end{array}; z^2\right],$$

and holds for $\beta+2\mu > 0$, and $\beta-2\rho < 1$. Setting $\beta = 2$, $z = \frac{1}{\alpha}$, $\mu = m+\frac{d-1}{2}$ and $\rho = \nu+\frac{d+1}{2}$ (where $\nu \notin \mathbb{Z}_+$) yields

$$\int_0^\infty \frac{t J_{m+\frac{d-1}{2}}^2(t)}{\left(\frac{1}{\alpha^2}+t^2\right)^{\nu+\frac{d+1}{2}}} dt$$

$$= \frac{1}{2^{2\nu+d}} \frac{\Gamma(m-\nu)\Gamma(2\nu+d)}{\Gamma\left(\nu+\frac{d+1}{2}\right)^2 \Gamma(m+\nu+d)} {}_1F_2\left[\begin{array}{c} \nu+\frac{d}{2} \\ \nu+1-m\ \ m+\nu+d \end{array}; \frac{1}{\alpha^2}\right]$$

$$+ \frac{1}{(2\alpha)^{2m}} \frac{\alpha^{2\nu}}{2^d} \frac{\Gamma(\nu-m)}{\Gamma\left(\nu+\frac{d+1}{2}\right)\Gamma\left(m+\frac{d+1}{2}\right)} {}_1F_2\left[\begin{array}{c} m+\frac{d}{2} \\ m-\nu+1\ \ 2m+d \end{array}; \frac{1}{\alpha^2}\right].$$

We remark that the ${}_2F_3$ hypergeometric functions from the formula collapse to ${}_1F_2$ hypergeometric functions in the application above, this is due to a repeated parameter appearing in both case; $\nu+\frac{d+1}{2}$ in the first instance and $m+\frac{d+1}{2}$ in the second. With this integral computed we can conclude that

$$b_{m,d,\mathcal{M}}(\boldsymbol{\lambda}) = \frac{2\pi^{\frac{d+1}{2}}}{\Gamma(\nu)(2\alpha)^{2\nu}} \frac{\Gamma(2\nu+d)}{\Gamma\left(\nu+\frac{d+1}{2}\right)} \frac{\Gamma(m-\nu)\kappa_{n,d}}{\Gamma(m+\nu+d)} {}_1F_2\left[\begin{array}{c} \nu+\frac{d}{2} \\ \nu+1-m\ \ m+\nu+d \end{array}; \frac{1}{\alpha^2}\right]$$

$$+ \frac{2\pi^{\frac{d+1}{2}}}{\Gamma(\nu)} \frac{\Gamma(\nu-m)}{\Gamma\left(m+\frac{d+1}{2}\right)} \frac{\kappa_{m,d}}{(2\alpha)^m} {}_1F_2\left[\begin{array}{c} m+\frac{d}{2} \\ m-\nu+1\ \ 2m+d \end{array}; \frac{1}{\alpha^2}\right].$$

Applying (23) we can write this as

$$b_{m,d,\mathcal{M}}(\boldsymbol{\lambda}) = (2\pi)^{\frac{d}{2}} \frac{2^{\frac{d}{2}}\Gamma\left(\nu+\frac{d}{2}\right)}{\Gamma(\nu)\alpha^{2\nu}} \frac{\Gamma(m-\nu)\kappa_{m,d}}{\Gamma(m+\nu+d)} {}_1F_2\left[\begin{array}{c} \nu+\frac{d}{2} \\ \nu+1-m\ \ m+\nu+d \end{array}; \frac{1}{\alpha^2}\right]$$

$$+ \frac{2\pi^{\frac{d+3}{2}}}{\Gamma(\nu)} \frac{(-1)^m\kappa_{m,d}}{\Gamma(m+1-\nu)\Gamma\left(m+\frac{d+1}{2}\right)(2\alpha)^m} {}_1F_2\left[\begin{array}{c} m+\frac{d}{2} \\ m-\nu+1\ \ 2m+d \end{array}; \frac{1}{\alpha^2}\right],$$

as required. $\qquad\qquad\square$

**Proposition 5.2 Statements 2 and 3.**

*Proof.* The following result provides the large parameter asymptotic behaviour of a ${}_1F_2$ of the same style as the first term in (30), it is adapted from Luke (1969) 7.3(11)

$${}_1F_2\left[\begin{array}{c} a \\ b-m\ \ c+m \end{array}; z\right] = 1 + \sum_{j=1}^n \frac{(a)_j z^j}{(b-m)_j(c+m)_j j!} + O\left(\frac{1}{m^{2n+2}}\right), \tag{41}$$

where $m-b \neq 0, 1, 2 \dots$. The next result is adapted from Luke (1969) 7.3(8) and provides the large parameter asymptotic behaviour of a ${}_1F_2$ of the same style as the second term in (30)

$$\,_1F_2\left[\begin{matrix}\alpha+m\\\beta+m\ \ 2m+\lambda+1\end{matrix};z\right] = 1 + \sum_{j=1}^{n}\frac{(\alpha+m)_j z^j}{(\beta+m)_j(2m+\lambda+1)_j j!} + O\left(\frac{1}{m^n}\right). \tag{42}$$

Applying Stirling's formula (11) we can deduce that the constant $\kappa_{m,d}$ (27) grows asymptotically as

$$\kappa_{m,d} \sim \frac{m^{d-1}}{2^{d-1}\pi^{\frac{d}{2}}\Gamma\left(\frac{d}{2}\right)}. \tag{43}$$

In addition, Stirling's formula also gives the following asymptotics for the Gamma functions involving $m$ appearing in (30)

$$\frac{\Gamma(m-\nu)}{\Gamma(m+\nu+d)} \sim \frac{1}{m^{2\nu+d}} \quad \text{and} \quad \frac{1}{\Gamma(m+1-\nu)\Gamma\left(m+\frac{d+1}{2}\right)} \sim \frac{1}{2\pi}\frac{1}{m^{\frac{d+1}{2}-\nu}}\left(\frac{e}{m}\right)^{2m}. \tag{44}$$

Using these asymptotic components in (30) we can deduce that, for large $m$, we have

$$b_{m,d,\mathcal{M}}(\boldsymbol{\lambda}) \sim \frac{2}{\alpha^{2\nu}}\frac{\Gamma\left(\nu+\frac{d}{2}\right)}{\Gamma(\nu)\Gamma\left(\frac{d}{2}\right)}\frac{1}{m^{1+2\nu}}\left[1+O\left(\frac{1}{m^2}\right)\right] + \frac{\sqrt{\pi}}{2^{d-1}}\frac{(-1)^m m^{\nu-\frac{d}{2}}}{\Gamma\left(\frac{d}{2}\right)}\left(\frac{e^2}{2\alpha m^2}\right)^m\left[1+O\left(\frac{1}{m}\right)\right].$$

Clearly the second component of the above asymptotic decays at an exponentially fast rate and so, to leading order, we have

$$b_{m,d,\mathcal{M}}(\boldsymbol{\lambda}) \sim \frac{2}{\alpha^{2\nu}}\frac{\Gamma\left(\nu+\frac{d}{2}\right)}{\Gamma(\nu)\Gamma\left(\frac{d}{2}\right)}\frac{1}{m^{1+2\nu}}.$$

Lemma 3.1 shows that the native space $\mathcal{N}_{\psi_{\mathcal{M}}}$ possesses the stated reproducing kernel properties. The norm equivalence of $\mathcal{N}_{\psi_{\mathcal{M}}}$ to the Sobolev space of order $\nu + \frac{d}{2}$ follows from Remark 3.1 and the established asymptotic decay rate of $b_{m,d,\mathcal{M}}(\boldsymbol{\lambda})$. $\qquad\square$

## B   Results Associated to the F Family

Here we present the proof of the 3 statements of Proposition 5.4 associated to the F family.

**Proof of Proposition 5.4 Statement 1**

*Proof.* For brevity we shall write $b_{m,d}$ for $b_{m,\mathcal{F}}(\boldsymbol{\lambda})$ in this proof. Applying (22) we have

$$b_{m,d} = \frac{B(\alpha,\nu+\tau)}{B(\alpha,\nu)}\frac{\sqrt{\pi}}{2^{m+d-2}\Gamma\left(\frac{d}{2}\right)}\frac{\Gamma(m+d-1)}{m!\Gamma\left(m+\frac{d-1}{2}\right)}\sum_{j=0}^{\infty}\frac{(\tau)_{m+2j}(\alpha)_{m+2j}}{(\alpha+\nu+\tau)_{m+2j}j!2^{2j}\left(m+\frac{d+1}{2}\right)_j}.$$

The following identities are taken from Prudnikov et al. (1981a) Appendix 1.6

$$(x)_{2j} = 2^{2j}\left(\frac{x}{2}\right)_j\left(\frac{x+1}{2}\right)_j \quad \text{and} \quad (x)_{m+2j} = (x)_m(x+m)_{2j}. \tag{45}$$

Applying these we can show that

$$\frac{B(\alpha,\nu+\tau)}{B(\alpha,\nu)}\frac{(\tau)_{m+2j}(\alpha)_{m+2j}}{(\alpha+\nu+\tau)_{m+2j}m!} = b_m(\tau,\alpha,\nu)\frac{2^{2j}\left(\frac{\alpha+m}{2}\right)_j\left(\frac{\alpha+m+1}{2}\right)_j\left(\frac{\tau+m}{2}\right)_j\left(\frac{\tau+m+1}{2}\right)_j}{\left(\frac{\alpha+\nu+\tau+m}{2}\right)_j\left(\frac{\alpha+\nu+\tau+m+1}{2}\right)_j}$$

and so

$$b_{m,d} = \frac{b_m(\tau,\alpha,\nu)}{2^{m+d-2}}\frac{\Gamma(m+d-1)}{\Gamma\left(m+\frac{d-1}{2}\right)}\frac{\sqrt{\pi}}{\Gamma\left(\frac{d}{2}\right)}\sum_{j=0}^{\infty}\frac{\left(\frac{\alpha+m}{2}\right)_j\left(\frac{\alpha+m+1}{2}\right)_j\left(\frac{\tau+m}{2}\right)_j\left(\frac{\tau+m+1}{2}\right)_j}{\left(\frac{\alpha+\nu+\tau+m}{2}\right)_j\left(\frac{\alpha+\nu+\tau+m+1}{2}\right)_j\left(m+\frac{d+1}{2}\right)_j j!}$$

$$= \frac{b_m(\tau,\alpha,\nu)}{2^{m+d-2}}\frac{\Gamma(m+d-1)}{\Gamma\left(m+\frac{d-1}{2}\right)}\frac{\sqrt{\pi}}{\Gamma\left(\frac{d}{2}\right)}\,_4F_3\left[\begin{matrix}\frac{\alpha+m}{2}\ \ \frac{\alpha+m+1}{2}\ \ \frac{\tau+m}{2}\ \ \frac{\tau+m+1}{2}\\\frac{\alpha+\nu+\tau+m}{2}\ \ \frac{\alpha+\nu+\tau+m+1}{2}\ \ m+\frac{d+1}{2}\end{matrix};1\right],$$

where, in the final line, we recognise the infinite series as the $_4F_3$ hypergeometric function.  □

**Proposition 5.4 Statements 2 and 3.**

*Proof.* We begin by examining the asymptotic decay of the multiple of the $_4F_3$ hypergeometric function from (33), i.e., we consider

$$C_{m,d}(\tau, \alpha, \nu) = \frac{\sqrt{\pi}}{\Gamma\left(\frac{d}{2}\right) 2^{m+d-2}} \frac{\Gamma(m+d-1)}{\Gamma\left(m+\frac{d-1}{2}\right)} \frac{B(\alpha, \nu+\tau)}{B(\alpha, \nu)} \frac{(\tau)_m (\alpha)_m}{(\alpha+\nu+\tau)_m m!}$$

$$= \frac{\Gamma(\nu+\alpha)\Gamma(\nu+\tau)}{\Gamma(\alpha)\Gamma(\nu)\Gamma(\tau)} \frac{\sqrt{\pi}}{\Gamma\left(\frac{d}{2}\right) 2^{m+d-2}} \frac{\Gamma(m+d-1)}{\Gamma\left(m+\frac{d-1}{2}\right)} \frac{\Gamma(m+\tau)\Gamma(m+\alpha)}{\Gamma(m+\alpha+\nu+\tau)\Gamma(m+1)}.$$

In the case where $m$ is large we can apply Stirling's asymptotic formula (11) to deduce that

$$C_{m,d}(\tau, \alpha, \nu) \sim \frac{\Gamma(\nu+\alpha)\Gamma(\nu+\tau)}{\Gamma(\alpha)\Gamma(\nu)\Gamma(\tau)} \frac{\sqrt{\pi}}{\Gamma\left(\frac{d}{2}\right) 2^{n+d-2}} \frac{n^{\frac{d-1}{2}}}{m^{1+\nu}}. \tag{46}$$

We now move on to the asymptotic decay of the $_4F_3$ hypergeometric function from (33). The following formula is taken from Prudnikov et al. (1983) 7.2.3(9)

$$_{p+1}F_{q+1}\left[\begin{matrix} \beta & \alpha_p \\ \beta+\sigma & \rho_q \end{matrix}; z\right] = \frac{\Gamma(\beta+\sigma)}{\Gamma(\beta)\Gamma(\sigma)} \int_0^1 t^{\beta-1}(1-t)^{\sigma-1} {}_pF_q\left[\begin{matrix} \alpha_p \\ \rho_q \end{matrix}; zt\right] dt.$$

Applying this to the $_4F_3$ hypergeometric function from (33), with $z = 1$, $\beta = \frac{\alpha+m+1}{2}$ and $\sigma = \frac{m+d-\alpha}{2}$, (such that $\beta+\sigma = m+\frac{d+1}{2}$) we have that

$$_4F_3\left[\begin{matrix} \frac{\alpha+m+1}{2} & \frac{\alpha+m}{2} & \frac{\tau+m}{2} & \frac{\tau+m+1}{2} \\ m+\frac{d+1}{2} & \frac{\alpha+\nu+\tau+m}{2} & \frac{\alpha+\nu+\tau+m+1}{2} \end{matrix}; 1\right]$$

$$= \frac{\Gamma\left(m+\frac{d+1}{2}\right)}{\Gamma\left(\frac{m+\alpha+1}{2}\right)\Gamma\left(\frac{m+d-\alpha}{2}\right)} \int_0^1 t^{\frac{m+\alpha-1}{2}}(1-t)^{\frac{m-\alpha+d-2}{2}} {}_3F_2\left[\begin{matrix} \frac{\alpha}{2}+\frac{m}{2} & \frac{\tau}{2}+\frac{m}{2} & \frac{\tau+1}{2}+\frac{m}{2} \\ \frac{\alpha+\nu+\tau}{2}+\frac{m}{2} & \frac{\alpha+\nu+\tau+1}{2}+\frac{m}{2} \end{matrix}; t\right] dt \tag{47}$$

The following identity is taken from Luke (1969) 7.3(3)

$$_{p+1}F_p\left[\begin{matrix} a_{p+1}+r \\ b_p+r \end{matrix}; t\right] = (1-t)^\xi \left[1 + \frac{d_1 t}{2r} + \sum_{k=2}^n \frac{d_k}{r^k} + O\left(\frac{1}{r^{n+1}}\right)\right],$$

$$\text{where} \quad \xi = \sum_{j=1}^p b_j - \sum_{j=1}^{p+1} a_j - r, \qquad d_1 = (\xi+r)^2 + \sum_{j=1}^p b_j^2 - \sum_{j=1}^{p+1} a_j^2, \tag{48}$$

$$d_k = \sum_{s=1}^k \beta_{k,s} t^s, \quad (2 \le k \le m), \quad \text{and} \qquad |\arg(1-t)| < \pi.$$

The quantities $\beta_{k,s}$ above depend only on the parameters of $a_{p+1}$ and $b_p$. For $p = 2$ we can use (48) to write the $_3F_2$ hypergeometric function appearing in the integral (47) as follows

$$_3F_2\left[\begin{matrix} \frac{\alpha}{2}+\frac{m}{2} & \frac{\tau}{2}+\frac{m}{2} & \frac{\tau+1}{2}+\frac{m}{2} \\ \frac{\alpha+\nu+\tau}{2}+\frac{m}{2} & \frac{\alpha+\nu+\tau+1}{2}+\frac{m}{2} \end{matrix}; t\right] = (1-t)^{\frac{\alpha-m}{2}+\nu}\left[1 + \frac{d_1 t}{m} + \frac{\beta_{2,1}t+\beta_{2,2}t^2}{\left(\frac{m}{2}\right)^2} + O\left(\frac{1}{m^3}\right)\right].$$

We can use the above to write

$$_4F_3\left[\begin{matrix} \frac{\alpha+m+1}{2} & \frac{\alpha+m}{2} & \frac{\tau+m}{2} & \frac{\tau+m+1}{2} \\ m+\frac{d+1}{2} & \frac{\alpha+\nu+\tau+m}{2} & \frac{\alpha+\nu+\tau+m+1}{2} \end{matrix}; 1\right]$$

$$= \frac{\Gamma\left(m+\frac{d+1}{2}\right)}{\Gamma\left(\frac{m+\alpha+1}{2}\right)\Gamma\left(\frac{m+d-\alpha}{2}\right)}\left[I_{m,d,\nu,\alpha}(0) + \left(\frac{d_1}{m}+\frac{4\beta_{1,2}}{m^2}\right)I_{m,d,\nu,\alpha}(1) + \frac{4\beta_{2,2}}{m^2}I_{m,d,\nu,\alpha}(2) + O\left(\frac{1}{m^3}\right)\right], \tag{49}$$

where

$$I_{m,d,\nu,\alpha}(j) = \int_0^1 t^{\frac{m+\alpha-1}{2}+j}(1-t)^{\frac{d-2}{2}+\nu}dt$$

$$= B\left(\frac{m}{2} + \frac{\alpha+1}{2} + j, \frac{d}{2} + \nu\right) = \frac{\Gamma\left(\frac{m}{2} + \frac{\alpha+1}{2} + j\right)\Gamma\left(\frac{d}{2} + \nu\right)}{\Gamma\left(\frac{m}{2} + \frac{\alpha+1+d}{2} + j + \nu\right)}, \quad j = 0, 1, 2.$$

In the case where $m$ is large we can apply Stirling's asymptotic formula (11) to deduce that

$$I_{m,d,\nu,\alpha}(j) \sim \frac{\Gamma\left(\frac{d}{2} + \nu\right)2^{\frac{d}{2}+\nu}}{m^{\frac{d}{2}+\nu}}, \quad j = 0, 1, 2, \quad \text{and} \quad \frac{\Gamma\left(m + \frac{d+1}{2}\right)}{\Gamma\left(\frac{m+\alpha+1}{2}\right)\Gamma\left(\frac{m+d-\alpha}{2}\right)} \sim \frac{2^{\frac{d-1}{2}}}{\sqrt{2\pi}}2^m\sqrt{m}.$$

These two results allow us to deduce that, when $n$ is large, we have the following asymptotic formula

$$\sideset{_4}{_3}{\mathop{F}}\left[\begin{matrix}\frac{\alpha+m+1}{2} & \frac{\alpha+m}{2} & \frac{\tau+m}{2} & \frac{\tau+m+1}{2} \\ m+\frac{d+1}{2} & \frac{\alpha+\nu+\tau+m}{2} & \frac{\alpha+\nu+\tau+m+1}{2}\end{matrix}; 1\right] \sim \frac{\Gamma\left(\frac{d}{2}+\nu\right)}{2\sqrt{\pi}}\frac{2^{m+d+\nu}}{m^{\frac{d-1}{2}+\nu}}\left(1 + \frac{d_1}{m} + \frac{(\beta_{2,1}t + \beta_{2,2}t^2)}{m^2}\right). \qquad (50)$$

Bringing (50) and (46) together, we can conclude that

$$b_{m,d} \sim \frac{\Gamma(\nu+\alpha)\Gamma(\nu+\tau)}{\Gamma(\alpha)\Gamma(\nu)\Gamma(\tau)}\frac{2^{\nu+1}\Gamma\left(\frac{d}{2}+\nu\right)}{\Gamma\left(\frac{d}{2}\right)}\frac{1}{m^{1+2\nu}}.$$

Lemma 3.1 shows that the native space $\mathcal{N}_{\psi_{\mathcal{F}}}$ possesses the stated reproducing kernel properties. The norm equivalence of $\mathcal{N}_{\psi_{\mathcal{F}}}$ to the Sobolev space of order $\nu + \frac{d}{2}$ follows from Remark 3.1 and the established asymptotic decay rate of $b_{m,d,\mathcal{F}}(\boldsymbol{\lambda})$. □

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
