# OpenReview forum: "Sobolev Spaces, Kernels and Discrepancies over Hyperspheres"
_TMLR — Accepted by TMLR_

### Review · Reviewer_RopF · 2023-01-08

**Summary Of Contributions:**

This paper studies the properties of Hilbert spaces defined by certain kernels on sphere. Specifically, the conditions on equivalence between Hilbert spaces and Sobolev spaces are presented. By giving the explicit Fourier-Schoenberg sequences, this works closes the gap of kernels spaces equivalence assumptions in Barp etal. (2022).  Thus, it extends the studies of kernel methods from Euclidean spaces to non-Euclidean space.

**Audience:**

No

**Claims And Evidence:**

Yes

**Requested Changes:**

More discussions on the applications of kernel methods on sphere will be welcome.

**Strengths And Weaknesses:**

Strengths:
The paper is well organized.
The main contribution is to close the gap between thoery and practice of kernel cubature and kernel discrepancy on spheres by carefully deriving the mathematical expressions of Fourier-Schoenberg sequences for several  parametric classes of members $\psi$ .


Weaknesses:
I'm not familiar with the Riemann Stein's method. Perhaps the application of kernel methods on sphere in machine learning is limited, so some discussions on the real application will be helpful to broader readers.

Typos:
page 2: "The theory of kernel cubature on is well-developed", on what.

---

> ### Author Response · Authors · 2023-02-19
> **Author Response**
>
> Thank you for your helpful suggestion to spell out the wider context of our contribution, which was also echoed by Reviewer GKnX.
>
> Here we take the opportunity to describe some potential applications of our work in ML:
>
> - Kernels have been used for at least two decades to approximate the solution of (partial) differential equations (PDEs), and are particularly appealing in that they are "mesh free", allowing for arbitrary locations at which the gradient field is evaluated (see e.g. Fasshauer, 2007).  In these methods, the smoothness of the kernel should be selected to precisely match the smoothness of the solution to the PDE.  Our contribution characterises kernels that reproduce Sobolev spaces, and Sobolev spaces are also the most popular spaces in which PDEs are studied.  More recently, there has been growing interest within the ML community in probabilistic numerics (Hennig et al, 2022), in which kernel methods (Gaussian processes) are used to infer the solution of PDEs; this community represents an important target audience for our work (e.g. see Chen et al, 2021; Kramer et al, 2022; Pfortner et al, 2022).
>
> - In probabilistic ML, where computation of the Bayesian posterior distribution is often challenging, recent work on Stein's method (Gorham and Mackey, 2015) has inspired novel computational methodology based on the "Stein equation", an elliptic PDE whose solution space can again be cast as a Sobolev space in the case of S^d (Barp et al, 2022).  The associated kernel Stein discrepancy (Chwialkowski et al, 2016;  Liu et al, 2016) can be shown to control the weak convergence of measures on $\mathbb{S}^d$ whenever the kernel is a Sobolev kernel; our contribution extends the library of known Sobolev kernels that can be used in this context.  For other kernels on $\mathbb{S}^d$, to the best of our knowledge, no similar guarantees have been established.
>
> - In computer vision, the global illumination task is to render a glossy surface so as to simulate how an object would look in a natural environment.  Here, one calculates the brightness of a point on the object by integrating over all incoming light sources that may reflect and scatter off the object at that point - this is an integral over (the non-occluded half of) of $\mathbb{S}^d$, and for this kernel-based cubature methods have been proposed and studied (Marques et al, 2013, 2019, 2022).  Theoretical analysis of kernel cubature is arguably most natural in Sobolev spaces (Briol et al, 2019), and this community is another target audience for our work.
>
> - As a promising future audience for this work, we note that supervised learning tasks in which the state variable, or components thereof, live on $\mathbb{S}^d$ include gait and posture detection (where limbs are able to rotate at joints, and overall posture is to be inferred) and protein folding (where the angle of each bond between neighbouring amino acids is to be inferred).  Kernel methods are a promising but under-explored approach to tackle such problems, and our analysis can be used to inform the selection of a suitable kernel by considering what level of smoothness can be expected of the true regression function.  As a concrete example, human posture is usually encoded by a state variable in $\mathbb{S}^d \times \dots \times \mathbb{S}^d$, and since a tensor product of Sobolev kernels reproduces a Sobolev space with dominating mixed smoothness, our analysis of $\mathbb{S}^d$ sheds light also on the kernels required in this context.
>
> The relevance of our theoretical contribution to these applications of ML is now more clearly spelled out in Section 1 of the revised manuscript, and we believe that the paper has improved as a result.

---

### Review · Reviewer_c7Q3 · 2023-01-10

**Summary Of Contributions:**

This article studies the smoothness of reproducing kernel Hilbert spaces associated with geodesically isotropic kernels defined on the sphere. This study is relevant in the field of numerical integration. For instance, the rate of convergence of the worst-case error of integration of some kernel-based Cubature formulas depends on the smoothness of RKHS. The authors showed that the so-called ‘Fourier–Schoenberg sequence’ characterizes a geodesically isotropic kernel. This sequence is usually available for kernels defined on the infinite-dimensional sphere. The authors showed how to ‘project’ this sequence to finite-dimensional hyperspheres. Finally, they illustrated these results in the case of three families of kernels.



**Audience:**

Yes

**Claims And Evidence:**

Yes

**Requested Changes:**

I would recommend to give more insight on the relevance of geodesically isotropic kernels in the field of numerical integration.

Minor comments:

- After equation 3, it seems that it is enough to assume that $b_{m,d}$ is a summable sequence. Why do we need the assumption that it is a probability mass sequence?
- In equation 19: the notation is a bit confusing. The operator is defined on the set of sequences, not on the set of scalars.

**Strengths And Weaknesses:**

Overall the article is well-written and tackles an interesting question in the field of kernel-based cubature. Indeed, an RKHS is defined typically through the associated kernel, and it is not immediately clear how to characterize the smoothness of the functions belonging to such an RKHS based solely on the expression defining the kernel. In practice, one would compare the RKHS to classical functional spaces such as Sobolev spaces for which the smoothness is well-established. Characterizing the smoothness of RKHSs containing functions defined on the hypercube is a classic topic. Extending these results to hyperspheres was achieved for some dot product kernels [1]. In particular, in these works, the Fourier expansion through spherical harmonics is used instead of the classical Fourier expansion. This article extends these results to kernels defined through the geodesic distance. However, I identify two weaknesses in this work:
- The equivalence between some RKHSs and Sobolev spaces on the hyperspheres is already known in the literature [1]. In particular, cubature rules defined by two 'equivalent' kernels will achieve the same rate of convergence. I believe that the article does not emphasize enough on the benefits we may get from using geodesically isotropic kernels in this context and settles for a brief mention in the introduction ''... The statement is correct, albeit quite unnatural: constructive criticism in Porcu et al. (2018) and Porcu et al. (2021) show that direct constructions on the sphere would be preferable...''. It is preferable to give more insight on the relevance of this class of kernels.
- The arguments were illustrated for three families of kernels that are equivalent to the same class of functional spaces (Sobolev spaces), altoutgh they seem to be valid for a larger class of kernels. In particular, the study proposed in this work does not cover RKHSs of infinitely smooth functions; see [2] for an example. It would be interesting to harness the potential of Proposition 4.1 by investigating the necessary and sufficient conditions on the Shoenberg sequence so that (12) is satisfied. Such a result would be applicable to a larger class of kernels.




[1] Brauchart, Johann, E. Saff, I. Sloan, and R. Womersley. "QMC designs: optimal order quasi Monte Carlo integration schemes on the sphere." Mathematics of computation 83, no. 290 (2014)

[2] Minh, Ha Quang, Partha Niyogi, and Yuan Yao. "Mercer’s theorem, feature maps, and smoothing." International Conference on Computational Learning Theory. Springer, Berlin, Heidelberg, (2006)

---

> ### Author Response · Authors · 2023-02-19
> **Author Response**
>
> Thank you for your positive comments and constructive suggestions, including of related work, which have enabled us to improve the manuscript:
>
> In Section 5 we now include a discussion of Brauchart et al (2014) and Minh et al (2006).  Specifically, from Brauchart et al (2014) we highlight the two specific kernels that the authors show to be reproducing for the Sobolev space of order 3/2 on $\mathbb{S}^2$. (The other generalised kernel proposed in this paper requires some further modification in order for it to be viewed as a reproducing kernel.)  This serves as useful background, but we note that these kernels involve fewer degrees of freedom than the classes that we consider, and are in that sense less well-suited to applications in ML, where estimation of kernel parameters is usually essential to ensuring good performance of a kernel method.
>
> We now also flag the fact that $d$-Schoenberg coefficients are available (as in Minh et al, 2006) for classes of infinitely smooth functions, and we agree that it would be possible to employ the projection operator on such classes in order to access the $d$-Schoenberg sequences - this is also now remarked on in Section 5 of the main text.  However, we did not develop this last direction in detail as infinitely smooth kernels are generally inappropriate for the applications that motivate this work (for example, cubature methods rarely assume infinite smoothness of the integrand).
>
> In retrospect, we appreciate and agree with your comment that the motivation for using geodesically isotropic kernels could have been presented in more detail.  We have now addressed this in the second paragraph of the Introduction.  Here we point out the intuitive fact that when implementing a kernel method on any compact Riemannian manifold, then one should work with the natural distance metric on the manifold.  Euclidean kernels are widely available and so one could, in theory, use such kernels and view $\mathbb{S}^2$ as set in the ambient Euclidean space $\mathbb{R}^{d+1}$ - we caution against this approach.  The drawbacks are discussed in the cited papers (Gneiting, 2013; Porcu et al, 2018) but, for the sphere, they boil down to a substantial deviation between the Euclidean and geodesic distances at larger distances - this argument is now included in the Introduction.  The use of an unnatural metric is likely to negatively affect, for example, the integration of functions on $\mathbb{S}^2$, since the functions we are integrating are often most naturally defined on $\mathbb{S}^2$.  This includes the case of computing illumination integrals in computer vision, for example, where we are integrating over incoming light sources from a natural environment.
>
> The sphere is a prototypical example of a non-Euclidean manifold, and of course the relationship between geodesic and Euclidean distances could be extremely pathological in general (e.g. if the manifold `curves back on itself', so that two points are close in the ambient Euclidean sense but distant according to geodesic distance on the manifold).  For the sphere, at least, we can circumvent the problem of Euclidean versus geodesic distances by employing the Euclidean inner product formula, which links Euclidean distance to the natural geodesic distance; this allows us to view a radial kernel on the sphere directly as a function of the natural geodesic distance.  In the paper we illustrate this approach by proposing the spherical Matern kernel and, with its definition in place, we are able to apply spherical harmonic analysis to provide its Sobolev assessment (this being one of the novel contributions of the paper).  Such a strategy will clearly not be available for other, more complicated, manifolds in general, and alternative approaches will be required.
>
> Thank you, finally, for the two minor comments that you raised:
>
> - We included the assumption that the $d$-Schoenberg sequence is a probability mass sequence because we have in mind that the kernel can be used as a (positive definite) correlation model for a random field on the sphere.
>
> - We have re-worded Remark 4.1 to emphasize that the projection operator acts pointwise on the coefficients $b_m$ from the Schoenberg sequence.

---

### Review · Reviewer_GKnX · 2023-02-07

**Summary Of Contributions:**

The article is of theoretical nature and it studies RKHSs produced by geodesically isotropic kernels defined over spheres.

**Audience:**

Yes

**Claims And Evidence:**

No

**Requested Changes:**

Because of my appreciation of the paper presented above, it is difficult to suggest specific changes.

I would say that the main point to resolve is to thoroughly explain the impact of the authors' work in the ML community, hopefully through discussion, illustrations and experiments.

**Strengths And Weaknesses:**

In particular, the authors focus on "Sobolev-type kernels", a concept that to the best of this Reviewer's knowledge, is not widely known in the ML community (not even in the kernel community I would say).

Though I have worked with kernels, and understand the RKHS formulation and its role in kernel methods, I had a tough time understanding the objective, the contribution and the novelty of this work. I list a few comments below to support my recommendation

- Reading through the paper, and even though the authors list their contributions in the Introduction, it is hard to identify which parts of the manuscript are still background material and which are novel definitions or results. Perhaps the contributions are Prop 4.1, Lemma 5.1, Prop 5.3-5.4? It seems so but I couldn't confirm it.

- In general, I understand the the authors proved properties for some kernels/spaces, and even connected with existing kernels. I would say that the source of my confusion is that the value of these properties is not explained. Why is it important, for instance, to identify the Sobolev space associated to a given kernel? The authors state that a consequence of this property is its use on kernel cubature, however, there is not evidence shown by the authors of a practical advantage of their proposal

- I would say that the theoretical value of the paper is not clear for the  wider readership of TMLR. If the paper had applications perhaps it would reach a more general audience.

- I would like to clarify that I am not saying that every paper has to have experiments, but in this case the theoretical contribution has not been clearly presented.

- Another piece of evidence that this paper might appeal to a more theoretical audience and it is out of scope in TMLS is that it contains limited references of ML venues. Aside from a couple of IEEE and ICML references, most of them are from applied maths journals. Again, this is not a disadvantage but reveals that the readership of TMLR might not identify the contributions of this work.

Minor: typo on $\mathbb{S}^\infty$ in second line of Section 2

---

> ### Author Response · Authors · 2023-02-19
> **Author Response**
>
> Thank you for your thoughtful assessment.  In retrospect, we agree with you that our novel contributions were insufficiently "landmarked" in the manuscript, and that the importance of Sobolev characterisations could be more clearly explained.  In this revision (which uses red font to indicate improvements to the manuscript) we have taken significant steps to improve these two aspects of the manuscript:
>
> Landmarking of Novel Contributions:
>
> In Section 1.2 we have clarified which results are novel contributions of the manuscript - namely, establishing that the Matern and F classes of kernels on $\mathbb{S}^d$ are Sobolev kernels (Propositions 5.2 and 5.4 in the revised manuscript).  To the best of the authors’ knowledge the application of the projection operator (Proposition 4.1 in the revised manuscript) to derive $d$-Schoenberg sequences from $\mathbb{S}^\infty$ Schoenberg sequences is new, we outline this in the revision as well. On the other hand, the Fourier transform of the Matern kernel is well-known and, as such, we have replaced its proof with an appropriate reference instead.
>
> Importance of Sobolev Characterisations in ML:
>
> Thank you for the opportunity to expand on this point, which was also raised by Reviewer RopF.  To recap, a kernel on $\mathbb{S}^d$ can be associated to a reproducing kernel Hilbert space (RKHS) whose elements are functions from $\mathbb{S}^d$ to $\mathbb{R}$.  For "generic" learning tasks, such as support vector machine classification, the mathematical properties of the RKHS are not an especially important consideration, and one may select a suitable kernel based simply on minimisation of a training loss / evaluation on a test dataset.  On the other hand, there are specific applications where theoretical understanding of the RKHS is critical.
>
> One such example is the use of kernels to approximate the solution of (partial) differential equations (PDEs), where the solution space of the most commonly encountered elliptical PDEs are Sobolev spaces.  The use of a Sobolev kernel then ensures that the solution of the PDE belongs to the RKHS and, through the use of an appropriate kernel method, can be consistently approximated (see Fuselier and Wright, 2009, 2012; Hubbert et al, 2015).  The recent interest within the machine learning community in probabilistic numerics (Hennig et al, 2022), in which kernel methods (Gaussian processes) are used to infer the solution of PDEs, provides one important target audience for this work (e.g. see Chen et al, 2021; Kramer et al, 2022; Pfortner et al, 2022).
>
> A second important example comes from probabilistic ML, where computation of the Bayesian posterior distribution can often be difficult.  Recent work on Stein's method for Bayesian computation (Gorham and Mackey, 2015) centres on the so-called Stein equation, an elliptic PDE whose solution space can again be cast as a Sobolev space in the case of distributions defined on closed manifolds such as $\mathbb{S}^d$ (Barp et al, 2022).  The associated kernel Stein discrepancy (Chwialkowski et al, 2016;  Liu et al, 2016) can be shown to control the convergence of measures in distribution on $\mathbb{S}^d$ whenever the kernel is a Sobolev kernel.  For other kernels on $\mathbb{S}^d$, to the best of our knowledge, no similar guarantees have been established.  The community developing computational methodology based on the Stein equation and kernel Stein discrepancy are therefore a second target audience for this work.
>
> Section 1 of the revised manuscript has been re-written to more strongly emphasise the relevance of Sobolev kernels to these sub-communities within ML.  We hope that the theoretical value of our work is now clearly set into context.
>
> Finally, thank you for pointing out the typo in Section 2, which has been fixed.

---

### Decision · Action_Editors · 2023-03-27

**Recommendation:** Accept with minor revision

**Comment:**

The paper derives theoretical properties of reproducing kernel Hilbert spaces (RKHS) associated with geodesically isotropic kernels on $\mathbb{S}^{d}$. In particular, it gives a characterization of the smoothness of such RKHS. More precisely, it gives some conditions which implies that a RKHS with geodesically isotropic kernel belongs to the Sobolev space of $\mathbb{S}^d$ of order $\gamma>0$. These results are illustrated through three classes of isotropic kernels.
Although the paper does not really much detail on potential applications to ML, in my opinion, this may be of interest for further studies and may find natural applications for data defined on hyperspheres.

Before final acceptance, the authors should proofread their paper. I found a few typos. In particular,

- Eq (16) and other places, $S^{d-1}$
- p3 $ \sigma^2 \psi(\theta(\xi,\eta)$


**Audience:**

There was a slight difference in reviewers opinion on the relevance of the paper to TMLR audience.
I follow the majority and while there is no direct and natural connection to ML applications as such, the analysis developed in this work may be definitively of interest for the community and find applications for data defined on hyperspheres.

**Claims And Evidence:**

The contributions of this paper are well presented and from my understanding the results and their proof seem right.